# The Theodicy Challenge and the Intelligibility of the World

**Michał Oleksowicz** [1],* and **Michał Kłosowski** [2]

1 Institute of Philosophy, Nicolaus Copernicus University in Toruń, 87-100 Toruń, Poland
2 Faculty of Theology, Nicolaus Copernicus University in Toruń, 87-100 Toruń, Poland; klosek995@o2.pl
* Correspondence: michaloleksowicz@umk.pl

**Abstract:** This paper revisits one of the most difficult theological issues, namely God's infinite goodness and the presence of pain and suffering in the natural world. We deepen the understanding of this problem by referring to the philosophical notion of the intelligibility of the world. We argue that pain and suffering are present in biological evolution as a "structural necessity" for the development of more complex structures from simpler ones. The struggle for existence works as a necessary condition for the development of a sophisticated order of nature at the expense of an increase in pain and suffering. If this is so, arguments justifying the ways of a good, omniscient, and omnipotent God in a world where evil and suffering are widespread seem to be undercut. Therefore, we argue that the astonishing intelligibility of nature may help to open our understanding of whatever nature may reveal of itself. This notion—analyzed from ontic and epistemic perspectives—seems to be crucial in reflecting the evolving world, not only from the scientific point of view but also from the theological one.

**Keywords:** intelligibility; evolution; pain; evil; theodicy; religion-science

## 1. Introduction

Undoubtedly the publication in 1859 of Charles Darwin's "On the Origin of Species" was a landmark in the history of science comparable to the publication of Newton's "Principia". Darwin's idea that evolution is guided by natural selection was such an influential and innovative explanatory approach within science that it provoked major developments in the life sciences and at the same time launched a new worldview—the evolutionary worldview. The latter is fundamentally based on the conviction that the temporal character of natural processes, from the micro to the cosmological, is an essential feature. We live in a universe endowed with its own history. However, this history is not a purely positive one, for it also contains the reality of death, pain, and suffering. The latter elements, if viewed from the religious point of view, restate the classical theodicy question: if God is good, why is there evil, pain, and suffering?

It seems that common sense has already become largely accustomed to the fact that we live in the evolutionary world. Yet, the details of the evolutionary process and its philosophical or theological interpretation retain the power to astound, both scientifically and philosophically. Suffice it to say that science, notwithstanding its progress, does not have a satisfactory answer when it comes to the origins and nature of life (Davies 2004, pp. 93–120). In such a context, it is no surprise that the interaction between natural sciences, philosophy, and theology[1] lives on. In the mid-twentieth century, the dialogue between theology and science began to develop very dynamically. This was achieved not just by analyzing the interaction between both fields from a systematic and historical point of view (Russell 2002), it was also a result of the almost universal acceptance of the evolutionary image of the world. We do not aim to suggest that such an interaction is mainly due to the sort of explanatory gap within sciences which philosophy or theology tend to fill (e.g., Oleksowicz 2014). On the contrary, we want to emphasize that, in the context of

theodicy, the evolutionary worldview is a real challenge for both contemporary sciences and Christian theology.

For at least two decades, the origins of humanity and human morality have been analyzed from an evolutionary point of view, but equally religion itself and its manifestations, such as the thought patterns, rituals, symbols, and social attitudes generated by it, have become the subject of cognitive and evolutionary research (Richerson and Christiansen 2013; Szocik and van Eyghen 2021; Oleksowicz and Huzarek 2021). It seems that a good way to avoid what Alister McGrath (2001, pp. 64–70) calls the "essentialist error" in approaching the religion–science dialogue is not so much to identify a set of universal religious beliefs, common to all denominations or religions, or to treat science as a set of fixed and unchanging theories and methods, but rather to emphasize the core claims for a given religion and key scientific issues that are to be studied. It is for this reason that the subject of our attention is the problem of the intelligibility of the world, with reference to the theodicy issue, from both a philosophical and a theological perspective.

In this paper, we articulate notions of evil and evolution via the problem of the intelligibility of the world. In Section 2, we formulate the theodicy problem, and then in Section 2.1, we comment on philosophical and theological models that present an omniscient and omnipotent God in a world in which pain and suffering are widespread. In Section 2.2, we emphasize two main perspectives on the theodicy problem: the first is linked with the moral evil arising from human free will; the second is connected with physical or biological evil. In the rest of the paper, we concentrate on the latter issue. In Section 3, we start by formulating a preliminary definition of the concept of the intelligibility of the world and point out its ontic (intelligible reality) and epistemic (intelligent inquiry) aspects. Our discussion of the classical and modern attitude of mind in case of intelligibility in Section 3.1 shows how this notion intersects ontological and epistemological problems. In Sections 3.2 and 3.3, we argue that the issue of intelligibility may also shed further light on how we understand the evolutionary world and the problem of evil. In Section 3.4, we open the path for seeking a deeper intelligibility of the world. Finally, we conclude with a brief reflection on the fruitful role of the concept of intelligibility in developing a dialogue between science and religion.

## 2. If God Is Perfectly Good, What of Evil?

For centuries, many thinkers struggled with the problem apparently first formulated by Epicurus (341–270 BC) and quoted by Lactantius as follows:

> God either wishes to take away evils, and is unable; or He is able, and is unwilling; or He is neither willing nor able, or He is both willing and able. If He is willing and is unable, He is feeble, which is not in accordance with the character of God; if He is able and unwilling, He is envious, which is equally at variance with God; if He is neither willing nor able, He is both envious and feeble, and therefore not God; if He is both willing and able, which alone is suitable to God, from what source then are evils? or why does He not remove them? (Quoted in Hick 1966, p. 5)

We can refer to this problem as the theodicy challenge. The term "theodicy" comes from two Greek words: "theos" (God) and "dikaioun" (justify). In other words, this problem refers to the search for arguments that justify the ways of a good, omniscient, and omnipotent God in a world where evil and suffering are widespread. It is worth pointing out that while Epicurus makes no mention of time in discussing the theodicy challenge, the Hebrew view (as evidenced in the Bible) always wrestles with related issues emphasizing the temporal aspects of God activities. It suggests that answers to the theodicy challenge involving extended processes of some sort are more coherent with the Judeo-Christian view than timeless solutions[2]. In the above quotation, Epicurus presents a spectrum of four ways of conceiving of God in the face of evil: namely a God willing and unable, a God neither willing nor able, a God able and unwilling, or a God both willing and able. Since the concept of God willing and unable is not in accordance with our basic intuitions associated

with such a concept, in the next section, we will take the remaining threefold distinction as the basis for discussing more up-to-date models of God's action in the evolutionary world.

More than a century before Darwin, Gottfried Wilhelm Leibniz devoted an entire treatise, "Theodicy", to the search for arguments explaining God's allowance for the existence of evil in the world. His conclusion was that our world, despite the evil, is still the best of all possible worlds. Darwin, for instance, in his letter of 22 May 1860 to his friend, the Harvard botanist Asa Gray, expressed his astonishment that so much suffering was required by God for the evolution of species (Darwin Correspondence Project n.d.). It seems that Darwin in his thinking deeply confronted God and the evolutionary world: on the one hand, a beneficent and omnipotent God; on the other, brute force, chance, and excessive misery in the world. In this context, it is worth emphasizing that it does not seem that evolution itself, conceived as a historical process, is really the origin of the theodicy question. It is rather the presence of gratuitous physical evil or the mere existence of a long period of time before the human era in which creatures suffered and presumably not as a result of human actions. Such a brute fact seems to be therefore both God's fault and somewhat pointless. One may also ask whether somehow Darwin's theology was influencing science or maybe the other way around[3]. While we leave this problem as an open issue, it seems that Darwin was attracted to evolution partly as a step towards a "hands-off" model of God. Let us now examine models of God's action in the evolutionary world.

*2.1. Three Theodicy Models*

There has been a variety of philosophical and theological models presenting omniscient and omnipotent God in a world where pain and suffering are widespread but let us take the following range of views to guide our discussion (Alexander 2018, pp. 221–28). These views range from the "hands-off" model (God neither willing nor able), via the "freedom" model (God able and unwilling), to the "book of nature" model (God both willing and able). Let us briefly comment on each model.

Within the category of the hands-off model, we may include ideas of a self-emptying God as presented by Hans Jonas or the process theology of A.N. Whitehead. Basically, this model emphasizes that God gives the creation existence and autonomous development. The World is like a giant experiment on the part of God. Moreover, this model implies that God's omniscience and omnipotence are questioned. As a consequence, on the one hand, such a God cannot be blamed for the evil and suffering, but, on the other hand, we remove the traditional understanding of God. In short, God is represented here as unwilling and unable to act in the face of evil.

In the middle of our spectrum, we find different views that insist on God's omnipotence but also on the fact that God has deliberately chosen to restrict his omnipotence, allowing all creatures to express their own freedom. Within this category, we may include the proposals of John Haught, speaking about the restricted omnipotence of God, or Arthur Peacocke or John Polkinghorne, who stress human free will and freedom of the universe. All these proposals tend to describe the meeting of two freedoms: on the one hand, God's self-imposed restriction (kenosis); on the other, the world gifted with its own freedom. In this model, pain and suffering do not seem to be the fault of God but rather of evolution, since freedom is bestowed upon the natural world. It seems that advocates of this view tend to formulate the sort of deistic view according to which God creates the laws of nature and then the world proceeds according to said laws. As Alexander (2018, pp. 224–25) rightly notes, there are at least three problems linked with this model. First, strictly speaking, the term "freedom" makes sense when referring to human decision-making. In the case of the material world, it should be kept in mind that this is an anthropomorphic way of thinking. Second, it is true that Christian doctrine describes God, particularly the reality of the Incarnation, from the kenosis point of view, but at the same time, it cannot be forgotten that the Bible very often refers to the joyful and powerful actions of God the Creator (e.g., Genesis 1). Third, an objection that pertains to both hands-off and freedom models is that they do not provide a real answer to the problem of the evil and suffering entailed by the

evolutionary process. In fact, in the first case, God is unwilling and unable to act in front of evil and suffering. In the second case, God is able but unwilling to act. However, in both models, God is responsible, respectively, for the designed laws of the world or the uncertain outcomes of the free process, so the theodicy question remains unanswered.

According to the third position, God is the Author of the Book of Nature, having created the world out of nothing and sustaining it in existence. Although the metaphor of the Book has undergone many transformations in Western thought (Blumenberg 1984), a place of honor, from a historical point of view, is saved for Raymond of Sabunde (Ramon Sibiuda, c.1385–1436), who extensively spoke of nature like a book, that is, like a place wherein the Creator's traces are identifiable (Conti 2004). This metaphor evokes the robust Trinitarian theism, according to which God the Creator and Redeemer is responsible for the World by providing the order of creation and operating via secondary causes that have their own causal efficacy. In this sense, the model takes the problem of evil and suffering seriously. Since there is physical or moral evil in the world, the Author of the Book may be blamed. From a historical point of view, the leading figures of the Scientific Revolution (e.g., Copernicus, Gassendi, Boyle, Newton, Descartes) wished to avoid a clash between theology and the new sciences by resorting to a view of the two Books, i.e., the Book of Nature and the Book of Scripture. However, subsequently the tensions did not disappear but continued to smolder under the ashes, chiefly because the doctrine of the two Books changed its appearance. The two Books (nature and revelation), traditionally conceived in a dialogical perspective since written by the same God, have come to be considered in isolation as separate texts (nature versus revelation). If God, the Author of the Book, is really omnipotent and omniscient, and his purposes are being fulfilled in the created world where death, pain, and suffering are present, then the discussion about theodicy really starts and mainly goes in two directions.

### 2.2. Theodicy: Two Perspectives

The first direction is linked to the moral evil arising from human free will. This may seem like an easy way to justify God's goodness since it is enough to argue for the free human decision-making process and make humans responsible for their actions. However, if one considers, for example, the following bitter confession of Ivan in Dostoevsky's "The Brothers Karamazov"(part II, book V, chapter 4), the problem of moral evil does not seem to be so easily settled:

> (. . .) I renounce the higher harmony altogether. It's not worth the tears of that one tortured child (. . .) do not want more suffering. And if the sufferings of children go to swell the sum of sufferings which was necessary to pay for truth, then I protest that the truth is not worth such a price. (. . .) Too high a price is asked for harmony! (Dostoevsky 2009, p. 301; our translation)

Although the moral evil can be conceived of as the direct result of humans' actions—the tears of a tortured child—nevertheless, its concreteness and dramatic power is something that deeply influences not only human existence but also the presumed image of God who created such people and who is neither able nor willing to defend those who are innocent. We will not go further into the problem of moral evil here, but it is worth noting the vast philosophical literature of the twentieth century, a time marked by totalitarianism, that sought to deal with the theodicy issue from an existential point of view (e.g., Hannah Arendt, Hans Jonas, Albert Camus, Józef Tischner, among others).

Interestingly, Ivan's words above also express the presumed link between the value of life and its costs. Ivan is arguing that although pain and suffering are a part of life, they are too high a price to pay for life's harmony. In other words, we cannot reconcile a good God with the gratuitous suffering of even a single creature. This is the second direction that we want to further pursue in this paper. We leave aside the case of moral evil and focus on the physical or biological evil seen from the perspective of the third model. In this case, one can take Leibniz's position and argue that we live in the best of all possible worlds. In other words, what seems to us as a sort of dissonance (death, suffering, pain, earthquakes,

hurricanes, disastrous fires, species extinction, etc.) is part of a larger, harmonious whole. We may consider at least two stipulations coming from contemporary biology that may help us to better ground this Leibnizian view.

First, we should not "demonize" natural selection. In the current scientific literature, the question "what is the nature of natural selection?" remains a hard one to answer. Some have argued that natural selection is a sort of force (Sober 1984), others that it is a purely statistical historical process (Matthen and Ariew 2002), or that it is a causal process operating at the population level (Millstein 2006). The so-called new mechanists also have entered this debate since R.A. Skipper and R.L. Millstein's paper (Skipper and Millstein 2005). In fact, natural selection seems to be like a mechanism, since it is productive of certain outcomes, brings about adaptations, and is composed of interacting entities, that is, populations of organisms (with varying traits) and activities (i.e., interaction of organisms with the environment). Although natural selection is not the sole mechanism of evolution (i.e., among others are enumerated random genetic drift, mutation, niche construction, and gene flow or migration), it is a very conservative "biological mechanism." It works as a rigorous filter to reduce the amount of genetic variation in a population. Francisco Ayala defines it in the following apt way:

> Natural selection is the differential reproduction of alternative genetic variations, determined by the fact that some variations are beneficial because they increase the probability that the organisms having them will live longer or be more fertile than organisms having alternative variations (Ayala 2007, p. 51).

Briefly put, why should natural selection be more problematic for theodicy than pain, suffering, death, or moral evil? Why should natural selection be considered as something in nature that is unsuitable for the functioning of the world or harmony of life and contrary to God's goodness? Natural selection is just as much a part of the world that we live in as black holes, elephants, and the Higgs boson (Sollereder 2019; Southgate 2008).

Second, in light of current life sciences, it seems that animal and human pain and suffering are necessary concomitants of the evolutionary process rather than "unfortunate accidents." When looking at the phenomenon of life, we note the copresence of positive and negative aspects, that is, what contributes to the life and well-being of organisms (e.g., mutations, cells, viruses, bacteria, etc.) may also significantly go wrong and cause death or reduce their well-being (e.g., the apoptosis that can be a root cause of cancer or harmful action of certain viruses). In fact, being alive, as far as carbon-based life is concerned, goes hand in hand with carbon-based death. No animal can derive all its energy solely from chemical elements; they are dependent on food chains in which molecules are synthesized in other organisms and then consumed by other organisms eating the former ones. In the case of humans endowed with huge brains and consciousness, the energetic costs of life are even higher. Along with this comes the issue of pain, which is an essential property of more developed organisms endowed with nervous systems. In this case, the general "biological law" is simple: as sentience increases, awareness of pain becomes greater. Therefore, the complexity of nervous system, the awareness of environment, the experience of pain—all these aspects are tied together. The cost of existence—pain, suffering, death—seems to be huge, but the other side of the coin is that a high price is paid for the huge value of life (Pabjan 2018; Schneider 2020). Even in the case of humans, the association of any conscious experience with nervous systems is widely regarded as mysterious; nevertheless, it seems to us that we also have independent evidence that many animals are capable of experiencing pain and thus we are justified in rejecting any neo-Cartesian explanation that denies that animals have this ability (Rose and Adams 1989). It is beyond the scope of the present paper to further discuss the problem of sentience.

From our reflection upon the theodicy challenge in light of contemporary life sciences, we may formulate two preliminary conclusions:

(1) Pain and suffering are present in biological evolution as a "structural necessity" (Peacocke 1971, p. 138); in other words, death, pain, and suffering are essential elements of a carbon-based world where life is present.

(2) God's purposeful actions can be best understood as realized not in spite of but through natural selection.

In the evolutionary world, as far as we acquire knowledge about it, the presence of physical evil, pain, death, and suffering turns out to be an ontic necessity. These features are part of ongoing natural processes. So, it makes a sense to ask "why" and "how" the world is as it is. The most difficult task is then to show how such a "structural necessity" or "nomic regularity" (Murray 2008) might justify the specific evil of animal suffering either as a necessary condition or as an unavoidable by-product of natural processes. It is not obvious that nomic regularity requires physical evil:

> There is certainly no intrinsic connection between the two. After all, a world could be utterly nomically regular and as complex as you please without having any sentient creatures at all. Such a world would be regular, and would lack natural evil altogether. As a result, there will be no way to argue that the intrinsic goodness of nomic regularity, taken on its own, is sufficient to explain animal suffering (Murray 2008, p. 150).

From the theological point of view, nomic regularity means that God is the Author of the Book of Nature, and the latter operates according to certain laws or nomic necessities. God admits the presence of elements that may seem to us harmful or not beneficial, since the overall order of Creation assumes their presence and role in the evolution of life. However, it makes sense to ask why and how the world is as it is. That is, even granting that God wanted all the beaks, hooves, claws, suckers, gills, wings, and so on, why make it so that they cause pain? Why bother with evolution at all instead of just fast-forwarding evolutionary history to the human era?

If we want to theologically defend the view that God's existence is logically compatible with the physical evils we find in the world, then the available strategy is, as Murray (2008, pp. 10–40) suggests, to argue that there are no good reasons to accept that there just is gratuitous evil in the world. The evidential problem states that the amount of animal suffering in the world provides compelling evidence that God does not exist and that theism is irrational. Murray argues that in the light of the evolutionary image of the world, we are rather justified in believing that animal suffering is not gratuitous. Animal pain and suffering can be conceived in terms of benefits to the animals themselves, since they seem to be necessary to preserve the integrity of sentient organisms. If this is so, then we are not justified in concluding that physical evil is evidence of God's direct responsibility or even God's nonexistence. As Murray (2008, pp. 130–92) notes, it can be argued that a universe which moves from chaos to order in a law-like way is intrinsically good, and that animal suffering is an unavoidable by-product of such a universe. The intrinsic goodness of such a universe outweighs the animal suffering it inevitably produces.

This sort of justification of God by appealing to the nomic necessity of pain and suffering, however, relies on the implicit scientific realist reading of the evolutionary process. In other words, we are taking a positive epistemic attitude towards the content of our best evolutionary theories and models and do not consider them as merely empirically adequate. Since there is no definite winner in the realist vs. antirealist debate on the character of scientific theories (Wray 2018), we stress that various theological answers to the theodicy issue rely on certain philosophical readings, which might be neither unique nor conclusive, of scientific theories.

## 3. Articulating the Evil and Evolution

In this section, we further deepen our analysis of the notion of intelligibility from both an ontic and an epistemic perspective. Although terms like the intelligibility, rationality, or comprehensibility of the world are not synonyms, for our purposes, we will treat them as "family resemblance" concepts. It seems that these concepts are connected by a series of overlapping similarities, while no one feature is definitive for all of them. In fact, they are often used to define each other.

The intelligibility of the world, as Heller (2008, pp. 38–40) rightly notes, can be conceived of as a pre-assumption of science, that is, as a kind of working philosophical hypothesis that science tacitly assumes and which can be confirmed, rejected, or modified as science develops. Therefore, if science studies the world and does so successfully, this would mean that the world can be comprehended. By the world, we mean not only all that is not God but the totality of being: non-human, human, already known, still unknown, etc. In this sense, the intelligibility of the world is a hypothetical assumption of science, to which we attribute greater certainty as scientific theories progress. Intelligibility of the world means that the world can be known and that we manage to know it quite effectively.

The notion of intelligibility should be analyzed from an ontic (intelligible reality) and an epistemic (intelligent inquiry) perspective. "Certainly, the universe as we know it is one in which being and knowing are mutually related and conditioned, intelligible reality and intelligent inquiry belong together" (Torrance 2001, p. 3). In other words, our knowledge of the world is determined not just by the nature of what we know but also by our intellectual activity of knowing it. The recognition of this dual aspect of intelligibility can have a salutary effect on us (Torrance 2001, p. 2). On the one hand, it may prevent us from making unwarranted claims about the objectivity of our knowledge, while, on the other, it also reminds us that what we know has a reality independent from our knowing of it. Let us dwell on this dual—ontic and epistemic—aspect.

### 3.1. Classical and Modern Attitude of Mind: Why This Distinction Matters

The notion of intelligibility points out that the world we know is one in which being and understanding are mutually related and conditioned. The major philosophical issue can be formulated in the following way: granted that the world exists and is comprehensible to us, is it comprehensible because somehow it is intrinsically intelligible or rather is it extrinsically intelligible? The first option means that there is some immanent rationality quite independent of us which is the foundation of its being understandable to us; the second option means that intelligibility is something that we construct out of our mental operations and impose upon the world. Briefly put, it is a contrast between finding and creating meaning. Is the intelligibility something that derives from the world, or does it rather derive from our mental operations? Torrance (2001, pp. 1–31) has labeled these two stances, respectively, as the classical and modern attitudes of mind.

Undoubtedly, the classical attitude owes much to the absorption of Greek thought through patristic theology and philosophy (Barbour 1997, pp. 199–203, 209–14; Hooykaas 2000, pp. 1–28; Pedersen 1990). Essentially, early Christian authors tried to work out a synthesis of Platonic, Aristotelian, and Stoic thought. On the one hand, Greek philosophy, with a special concern for cosmology and epistemology, was emphasizing that comprehension of the world and the limitations of human cognition go together. What is not limited, determined, and defined is, for the human mind, incomprehensible. On the other hand, in ancient Greece cosmogony and theogony were closely connected, e.g., Ionian philosophers looking at nature as a deity, Plato's demiurge and the importance of the supreme Idea of Good in platonic philosophy, or Aristotle's Prime Mover. In this Greek philosophical context, which often assumed that there is a God-given order, the first Christian thinkers started to argue from a typically Christian perspective that an incomprehensible God, transcending all human thought about Him, impregnated the world with a rational order that is intelligible to us. It was upon this intrinsic intelligibility that Christian philosophy and theology further developed the doctrine of the God Creator giving the basis for the unitary view of the created world and comprehensive scientific way of knowing such a world. Furthermore, the doctrine of the goodness of the world has been reinforced by that of the Incarnation of the eternal Logos (Son) of God. This has strengthened the conviction about the world's intrinsic intelligibility, but at the same time the Christian emphasis on God's transcendence started to alter the concept of intrinsic intelligibility. This was because Christian thought concentrated on a profound integration of logos and being through the historical event of Incarnation. The immediate result was not a static vision of God, like

the Supreme Good or Prime Mover, but a relational conception of God actively present in the creation. As a consequence, the dominant view was that intrinsic natural intelligibility relies deeply on a supernatural God, since the separation between God and the world has been overcome through the Incarnation.

One of the main representants of the classical attitude in theology is Thomas Aquinas. His view can be included within the third theodicy model discussed in Section 2.1. Aquinas considers intelligibility within the broader context of his theory of truth as congruence of understanding and reality (adaequatio intellectus et rei). One can object that this view of truth leaves unanswered the issue of whether understanding conforms to things or things conform to understanding. Aquinas' realist intention seems to demand the former. In fact, in his commentary to Aristotle's "Metaphysics", Aquinas considers the ordo intelligendi (Comment in Metaph., proemium) by referring to knowledge via causes, the difference between intellectual and sensual knowledge, and capacity of intellect to abstract from material conditions. What he seems to have in mind is in fact the basic link between being and knowing which allows him to argue that whatever there is can be understood (Summa Contra Gentiles II, 98). According to Aquinas, the concept of being (ens) is deeply linked with the concept of truth (verum). But knowing a being does not imply knowing it truly. Aquinas states that it would be impossible to know a being if it were unintelligible, but it is possible to know a being even if we are not aware of its intelligibility (Summa Theologiae I, q. 16 a. 3 ad 3). In other words, our understanding assumes the inner intelligibility of being, but not vice versa. Aquinas is justifying the possibility of knowing by linking the concept of inherent intelligibility of being with understanding itself as the place of truth. The Thomistic perspective on the intelligibility finds its metaphysical foundation also in reference to Aristotelian hylomorphism. For Aquinas, the intelligibility of being results from the form, since the latter is the metaphysical principle of knowledge (Super Sent., lib. 1 d. 35 q. 1 a. 1 co.). In fact, for Aquinas, "every form is a certain participation in the likeness of the divine being" (In Phys. I, lect. 15), since God is the Creator of substantial forms. Summing up, for Aquinas, on the one hand, there is a basic correlation between being and knowing, and on the other, there is a correlation between intelligibility of being and comprehending the being in a truthful way. The ultimate source of the rationality of the world is the Creator God.

In contrast to the classical attitude, the modern one assumes that it is we who, by our way of knowing, clothe the world. This view transfers the intelligibility to the human domain, since it is our mental operation that writes the concept of form and structure into nature. For instance, perhaps the clearest proponent of this attitude was Kant who mainly focused on establishing the conditions of epistemic objectivity through the transcendental method. Torrance rightly notes that "from this dominance of the mental over the non-mental world there arises the notion of instrumentalist science with its powerful tool the technological rationality" (2001, p. 17). According to the modern attitude, scientists do not focus only on formulating accounts of how things are in the world; they also, if not primarily, do practical things for certain aims. In fact, one of the main reasons behind the great prestige of science in modernity seems to be that it is associated with the mentality of homo faber, for whom practical efficacy is used as evidence for the truth of certain scientific theories or claims. Alongside intrinsic intelligibility, therefore, we have the modern attitude according to which science comprises an instrumental set of techniques that can be put to any sort of practical intervention (e.g., mechanical, genetic, computational, ecological) in the world. Perhaps the difficulties for the followers of the classical attitude are at least in part due to how science's instrumentality seems to have increasingly displaced part of that intrinsic intelligibility.

Why does this distinction between classical and modern attitudes towards intelligibility matter? Intelligibility, interpreted either intrinsically or extrinsically, ultimately ends up being an irreducible category bound up with some basic philosophical concepts like being, meaning, or truth. Something is intelligible, rational, and comprehensible since it makes

sense to us. What is intelligible is the self-evident; it is given to us. The unintelligible is simply the unspeakable; it is not given to us. It seems that, as Peter Dear argues,

> in the historical development of science, the awkward and unresolved tension between instrumentality and natural philosophy has yielded views of the universe that are dependent on particular human conceptions of what makes sense (2008, p. 14).

Although what makes sense is historically and conceptually bounded, and the scientific theories produced by humans' epistemic capacity change; nevertheless, the intelligible texture of reality persists:

> Over the centuries, human reason has grasped things from nature and given them a scientific form, taking contextual elements and shaping insights. Scientific rationality has evolved; at the same time, something intelligible has remained, and the awareness of our complex and fragile epistemic situation has grown (Marcacci and Oleksowicz 2023, p. 12).

Although science and our way of expressing intelligibility are changing, we are always working under the pressure of the world as "an identifiable datum which we must affirm as such in positive assertions about it" (Torrance 2001, p. 55). The ontic and epistemic perspectives on intelligibility help us to expand the ground of rationality. The question arises as to whether this expansion goes beyond the world itself and human knowing and cries out for some transcendent ground of rationality. Before we address this issue, we will first further articulate the notion of intelligibility with reference to the concepts of self-organization and teleology.

### 3.2. Intelligibility and Evolution

Mariano Artigas, the author of many publications on the issue of religion–science (Artigas 1992a, 1992b, 1998, 2017; Mancini 2014), has significantly contributed to the study of the intelligibility of the world. When trying to explain the concept of intelligibility, Artigas refers primarily to the concepts of order and organization. Although these words are often used interchangeably, as P. Davies (2004, pp. 72–92) notes, organization is a concept that refers primarily to the global properties of the system. In a certain range of conditions, there may be a threshold at which the predictability of the behavior of a given system may break down and a given system suddenly changes to a new state, which may have completely new, global properties. Such a transition is often accompanied by a change in spatial or temporal patterns of the given process. Systems that undergo such complex changes are called self-organizing (e.g., phase transition phenomenon in physics, dissipative structures in biology, or the Belousov–Zhabotinsky reaction in chemistry). The latter concept expresses the fact that processes have some kind of internal dynamics ("self") that produces some functional and complex effect ("organization") (Artigas 1998, pp. 195–203). Such a feature of various natural processes may "push" scientific research, then, to search for rational patterns that explain not only the self-organization of particular processes, but even the world's self-organization.

The issue of self-organization of natural processes, according to Artigas, is related to the issue of directionality, i.e., a specific tendency of development (Artigas 1998, pp. 203–20). Distinguishing different levels of teleology (e.g., stages of development, tendency) helps to avoid anthropomorphic (intentional) thinking about natural phenomena. For Artigas, directionality, whether in the animate or inanimate world, means the existence of a certain tendency to obtain certain goals. Artigas interestingly notes that the ubiquity of directionality of natural processes is a component of the self-organization of natural phenomena. The latter is possible provided that, in addition to the elementary nature of phenomena (components and their role in a given process), the world and phenomena occurring in it, they function as a comprehensive system. In the natural world there exists a kind of holism, i.e., the sort of unity of various elements that are organized in a specific way and because

of specific purposes. Holism, functionality, cooperation, goal-directed activities—these concepts well articulate the issue of the intelligibility of the world.

In the context of our concerns, a question about the intelligibility of the evolutionary process should also be posed. From a historical point of view, Darwin's theory represents a new conception of what it means to account for many features of organic nature. While pre-Darwinian philosophical or theological answers sought to identify some features of the world that gave evidence of an intelligent Creator, Darwin's proposal, as already mentioned, involved two central issues—evolution and natural selection. The first has meant that new species emerged through transformation from older species; the second was the natural mechanism to bring about changes in groups of organisms that would produce new species. Although Darwin introduced natural selection in his work, in reality he could not give a satisfactory explanation of this mechanism. Until the advent of gene theory and the development of the modern synthesis, the given account of systematic selection of particular traits was deeply flawed. Indeed, the very fact that the technical term retained the word "selection" meant that the old teleological sense of intelligent design still lurked in the background (Dear 2008, pp. 91–114). As we noted earlier, even in the current scientific literature, the question "what is the nature of natural selection?" remains one of the hardest to address. Without entering into further details, the process of evolution is based on the variation of genetic changes and working mechanisms of selection. What seems to be crucial for our aims is that the mechanism of evolution (consisting of random genetic drift, mutations, gene flow or migration, natural selection) is, among other factors, responsible for the production of self-organized living forms. Moreover, the evolutionary process seems to be, to a certain degree, cumulative, in the sense that its elements operate within a limited area of possibilities created by the already existing self-organization of life. The evolutionary process, in the light of contemporary science, can be considered as more intelligible now than it was for Darwin himself.

### 3.3. Evil, Evolution, and Intelligibility Confronted

The metaphysical quest lurking in the background of our analysis of intelligibility can be formulated as follows: "if something already exists, why does it exist this way and not otherwise?" This question does not simply express awe in the face of what exists (the fact of being) but can be reformulated as the question: "why does the evolutionary world, endowed with pain, death, and suffering, exist in this way?" In the light of what we have discussed above, the answers to this question may be set out in the following way: the world is rational (comprehensible, intelligible); it is goal-directed; it is self-organized. Since, as mentioned in Section 3.1, the notion of intelligibility (rationality) expresses the fact that something has a meaning or sense for us, let us tackle the above question and offer a broader picture of possible answers:

1. Agnostic position: we do not know whether the world is intelligible;
2. Surrender position: the search for intelligibility is not worth pursuing;
3. The world is intelligible:

   a. Intrinsic (ontic) intelligibility: there is intrinsic intelligible structure (organization, order) of the world;

   b. Extrinsic (epistemic) intelligibility: we construct intelligibility out of our mental operations and impose it upon the world;

   c. Co-dependence of intrinsic and extrinsic intelligibility: a sort of congruence between our cognitive capacities and the order of the world;

4. Deeper intelligibility: the world is intelligible to us but only to a certain degree.

Let us comment on each of these points. So far, we have mainly developed point 3, by emphasizing that intelligibility can be conceived of as stemming from both the world as it is and our epistemic attitude to the world. The fact that we effectively know the world and manipulate it works as something of a counterargument to point 1. In fact, our ability to acquire understanding shows that the initial assumption of intelligibility has been further

reinforced by the development of our scientific knowledge. Point 2 means that our pursuit of intelligibility is at the same time the moral stance, since our attitude towards intelligibility is constituted by human acts of reason and will. Finally, it is important to note that the comprehensibility of the world is basically combined with a sort of realist stance insofar as we tend to think that our comprehension of the world gives us genuine knowledge about the external world independent of any perceiving subject (point 3). Although the world—as point 3 suggests—is accessible to rational investigation, in practice it is grasped by our limited human minds, theories, models, etc. The factuality of intelligibility does not exclude that it can have a limited character, which is conveyed in point 4. However, even limited comprehensibility is worthy of wonder.

How might the issue of intelligibility shed further light on the problem of evil? The first benefit may come in case of revisiting, for instance, the privation theory of evil according to which evil consists of the absence of good (privatio boni debiti), where good may be conceived of as order, the law, good will, virtue, existence, etc. In the light of what we have discussed, the Augustinian account of evil as a privatio boni—a privation of original goodness—should be properly understood (Rosenberg 2018). The conventional reading of Augustine seems to suggest that the Fall had a direct impact on the natural order. Briefly put, "had Adam and Eve not eaten of the fruit, there would be no physical evil." However, "if the commonly presented view of Augustine's teaching on the impact of the fall on the cosmos is mistaken, then in Augustinian terms, it is wrong to describe violent physical and biological acts in nature as evil" (Rosenberg 2018, p. 240). The other way of interpreting Augustine, that we are sympathetic to, treats decay, death, suffering, etc., as essentially a part of any contingent being. It would mean that for him, decay is an acknowledgment that the creation is fundamentally limited and distinct from God and evil is a relation to corruption in human soul.

The second aspect to be added is the idea of the "unintelligibility of evil" (Stróżewski 2008, pp. 13–31). If intelligibility, as we noted, may be treated as expressing rationality, organization, comprehensibility, purposefulness, then evil seems to be something that opposes what we conceive of as meaningful and understandable (Życiński 2014, pp. 103–22). The force of the question unde malum, if referring to evolution, stems from the fact that we are looking for the reason of evil. If we look at physical evil—death, pain, suffering, struggling among species for survival—and at the same time ask for the reason for such states of affairs, we are in fact confronted with the problem of the intelligibility of evil. Let us state the two answers to this question. The first one—as seen in the (1) preliminary conclusion of Section 2.2—suggests that death, pain, and suffering is inscribed into the process of life (Schneider 2020). These elements are intelligible since they are part of the intelligibility of being itself, even if this rationality of the world may seem striking to us or exceed our capacity to comprehend every detail of the evolutionary process. The second answer—contrary to the previous one—suggests that death, pain, and suffering is something that opposes the intelligibility of the world as we conceive it and results in something irrational and out of order. In the light of what we have discussed, we opt for the first answer. It seems to us that the death, pain, and suffering should be treated as the part of the life process, or theologically speaking, of Creation. These realities should not then be evaluated as merely privatio, but rather as essential a part of any contingent being. Therefore, we think that the physical evil, as a part of the life process, should be understood in a positive sense. It is important to note that our (in)ability to comprehend death, pain, and suffering depends heavily on the preliminary understanding of being itself, or more specifically life. If we conceive of life only in a positive way—well-being, absence of pain, overcoming of death—then those realities turn out to be the corruptio.

We can express the alternative between the positive and negative aspects of the evolutionary process of life in terms often adopted by the scholastic tradition: quoad nos and quoad se. Apart from our understanding of the world (quoad nos), be it philosophical or scientific comprehension, there can be some deeper reason for such order, the structure of the world (quoad se). This depth (point 4 above), if considered from a theological point

of view, can be understood as the shift to the deeper, ultimate grounding of intelligibility. In other words, the evolutionary image of the world, in which are present death, pain, and suffering, can be conceived in a positive, but at the same time, limited (quoad nos) manner. In this way, science and philosophy are opening the path to the theological answer, to which we will now turn.

### 3.4. Towards Deeper Intelligibility

So far, we have discussed in the case of intelligibility the following dual fact: on the one hand, the openness of the structure of the world to our rational endeavor; on the other, the openness of our knowing to the comprehensible structure of the world. This is what many thinkers and scientists have referred to as the sense of religious awe in front of the comprehensibility of the world. For instance, Einstein explicitly stated that "the eternal mystery of the world is its comprehensibility," and that the very fact that the world is comprehensible "is a miracle" (Einstein 1960, p. 292). The intelligibility manifests itself so strongly that some people have no hesitation in using religious vocabulary to describe it. What happens in this sort of shift in looking at the problem of comprehensibility is that instead of considering the world in a flat way, we

> start to look at it in the multidimensional way in which the universe as a whole, and everything within it, are found to have a meaning through an immanent intelligibility that ranges beyond the universe to an ultimate ground in the transcendent and uncreated Rationality of God (Torrance 2001, p. 44).

From the theistic perspective, such a shift derives from the doctrine of the Creation and that of the Incarnation. Both doctrines emphasize that the rationality of God intersects contingent being and divine being. Furthermore, the ontic and epistemic intelligibility of the world opens a unifying perspective for the consideration of God's and nature's agency, as we suggested in the (2) mid-conclusion in Section 2.2. Artigas expresses this point in the following way:

> the role of natural agency is fully recognized and, at the same time, is seen as supported by a founding divine action that does not oppose nature but rather provides it with its ultimate foundation. This perspective stresses that God usually acts to respect and protect the natural capacities of His creatures, as He has provided them with great and marvelous potentialities so that they may cooperate with God's plans in a great variety of ways. These potentialities are never exhausted, so that new results can always be produced or expected. Now this is possible through the use of the knowledge provided by scientific progress (Artigas 1998, p. 250; our translation).

Artigas rightly notes that such a worldview, which undertakes philosophical reflection on the problem of intelligibility, can be compatible with theological thinking about divine action in the world. Adopting such a perspective does not impose, for example, a conflicting juxtaposition of the operation of God's rationality and evolution in the biological world (Peters 2018). God rationally rules the world, but this does not mean that nature will behave in a completely orderly manner consistent with our concept of rationality. In other words, God's rational plan indicates goal-directed action, but this ultimate goal, which is God himself, is not within the scope of our direct knowledge. Our rational participation in this goal lies in the fact that although "our attitudes of mind [are] unstable" (Sap 9:14) and "it is hard enough for us to work what is on earth" (Sap 9:16); nevertheless, by using our cognitive capacities, we come to know to some extent this world which is pleasing to God.

This sort of multidimensional view, combining both the naturalistic and the theological perspective, offers a crucial, additional ingredient: there is not only the ontic and epistemic aspects of the comprehensibility of the world, but there is also a self-transcendence of the notion of intelligibility that we construct. Torrance lucidly describes this awe of comprehensibility and our searching for deeper intelligibility:

we cannot rationally break off our relation to the intelligibility of the universe at some arbitrary point of our own choosing; otherwise it would be not existence or reality that we are determined to apprehend. This is why scientific inquiry cannot come to a halt at any point we want, but must go on questioning its questions in order to let reality disclose itself to science indefinitely. The capacity of man for this kind of indefinite, unlimited, unbounded inquiry represents that which from his side is correlated to the intelligibility that reaches out indefinitely beyond him and which cries out for, and manifests itself as capable of, explanation in relation to some transcendent source and ground of rationality (Torrance 2001, p. 56).

## 4. Conclusions

In this paper we have sought to deepen our understanding of evolution and evil in relation to the notion of the intelligibility of the world. We have argued that the struggle for existence in the domain of life seems to be a necessary condition for the development of a sophisticated order of nature at the expense of an increase in pain and suffering. We have claimed that focusing on the astonishing intelligibility of nature may help open our understanding to whatever nature may reveal of itself in its positive or negative aspects. Confronting the issue of evil, evolution, and intelligibility has helped us to note, on the one hand, that physical evil does not have to be considered as unintelligible, and, on the other, that it is possible that the world is intelligible to us only in a limited way.

We have attended to the intrinsic (ontic) and extrinsic (epistemic) dimensions of intelligibility. The first aspect means that there is some immanent rationality quite independent of us; the second aspect means that intelligibility is something that we construct out of our mental operations. The examined reality and knowledge turn out to be dynamically related to each other. This dynamic connection largely parallels the two basic features of scientific endeavor: to offer an understanding of how things are and to treat reality instrumentally. Nevertheless, both aspects should be considered under the umbrella of intelligibility—the how-understanding and instrumental-understanding. Since the intelligibility (comprehensibility) of the world remains a fundamental feature of both scientific and theological enterprises, we hope to have demonstrated here that it may play an important role in fostering and advancing the dialogue between science and religion.

**Author Contributions:** Conceptualization, M.O.; investigation, M.O.; resources, M.O. and M.K.; writing (original draft), M.O. and M.K. (M.K. contributed more to the presentation of Aquinas' and Artigas' views); writing (review and editing), M.O. and M.K. All authors have read and agreed to the published version of the manuscript.

**Funding:** This publication was made possible through the support of John Templeton Foundation: New Horizons for Science and Religion in Central and Eastern Europe grant. The opinions expressed in this publication are those of the authors and do not necessarily reflect the view of the John Templeton Foundation.

**Institutional Review Board Statement:** Not applicable.

**Informed Consent Statement:** Not applicable.

**Acknowledgments:** Special thanks go to Mariusz Szmajdziński who provided good conditions for writing the article.

**Conflicts of Interest:** The authors declare no conflict of interest.

## Notes

[1] By "theology" we mean "a study in which, along with other axioms, at least one sentence is assumed which belongs to a given Creed and which is not sustained by persons other than the believers of a given religion" (Bochenski 1965, p. 14). Theology is therefore an attempt to make intellectually understandable what are considered to be the truths of faith of a particular religion.

[2] We would like to thank an anonymous reviewer for raising this point.

[3] We would like to thank an anonymous reviewer for raising this point and referring us to Darwin's correspondence.

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
