# Peer review of "The Theodicy Challenge and the Intelligibility of the World"

_religions, doi:10.3390/rel14121513_

Round 1
Reviewer 1 Report
Comments and Suggestions for Authors
Comments on the Quality of English LanguageThere are periodic lapses in the quality of English. The article needs careful copy-editing by a native speaker.
Author Response
All detailed responses are in the file.

Reviewer 2 Report
Comments and Suggestions for Authors
This is a thoughtful and generously referenced submission on a topic of some importance. I think that there are some logical gaps that are worth noting. There are also some noteworthy omissions from the literature regarding animal pain and evolution in analytic philosophy of religion. So I think that more work should be done before publication.
A first question is whether evolution is really the origin of the problem, or is it rather the mere existence of a long period of time in which creatures suffer prior to, and therefore presumably not due to, human actions---a period that seems to be therefore both God’s fault and somewhat pointless. That creatures change over time, and do so naturally, is not essential to the argument. Thus old-Earth creationism already faced this problem: whether or not God specially creates from time to time, animals spend a long time eating each other and presumably suffering, and this is caused by God and not by Adam’s sin. Possibly one would find interesting thoughts about it decades earlier in the 19th century than the author, pointing to Darwin, leads us to expect. It is worthwhile noticing that one’s argument follows from logically weaker premises than one might accept.
Regarding Darwin, evolution and theodicy, it was worth recalling that Darwin was attracted to evolution partly as a step towards hands-off theodicy himself. Recall his remarks on parasitic wasps: “I cannot persuade myself that a beneficent & omnipotent God would have designedly created the Ichneumonidæ with the express intention of their feeding within the living bodies of caterpillars,” https://www.darwinproject.ac.uk/letter/DCP-LETT-2814.xml. Is Darwin’s theology influencing science? I wonder if the author thinks that Darwin made progress, regress, a lateral move, or something else regarding theodicy here? This is more a matter of curiosity than an objection or concern.
I appreciate the author’s and Denis Alexander’s critique of certain efforts to give the universe freedom.
Regarding God-of-the-gaps arguments, they have indeed been criticized, but they have also been defended in some instances perhaps unfamiliar to the author.
John Mark Reynolds, "God of the Gaps," in William Dembski, editor, _Mere Creation_. InterVarsity Press, Downers Grove (1998) pp. 313-331.
J. P. Moreland, “Science, miracles, agency theory & the God-of-the-gaps,” in R. Douglas Geivett and Gary R. Habermas (editors), _In Defense of Miracles_. InterVarsity Press, Downers Grove, (1997), pp. 132-148, 294-296.
David Snoke, “In Favor of God-of-the-Gaps Reasoning,” _Perspectives on Science & Christian Faith_ 53 (Sept. 2001), pages 152-158.
Robert Larmer, “Is there Anything Wrong with ‘God of the Gaps’ Reasoning?” _International Journal for Philosophy of Religion_ 52 (2002), pages 129-142.
Ronald G. Larson, “Revisiting the God of the Gaps,” _Perspectives on Science and Christian Faith_ 61 (March 2009), pages 13-22.
Here is a minor verbal point. I appreciate the author’s providing us Epicurus’s formulation of the problem of evil, but I would be reluctant to call something with more than two horns a “dilemma.” One can have a trilemma, for example.
It might be worth pointing out that Epicurus makes no mention of time, but Biblical wrestling with related issues (e.g., Habakkuk or Psalm 73) always emphasizes time: wait for God to act. That doesn’t necessarily solve the problem, but it suggests that answers involving extended processes of some sort are more coherent with the Judeo-Christian scriptures than are timeless solutions.
On p. 5, the author leaps over a difficult issue in the philosophy of mind: why should there be any conscious experience associated with nervous systems? This association even in humans is widely regarded as quite mysterious---David Chalmers, etc. If one were prepared to say (as few people are) that it is even false---think of Descartes on animals---then the problem would disappear. Animal-robots would still eat each other, bleat and die, but no pain would be felt. That is a weird idea to most people (including me), but one should not simply fail to imagine it given that consciousness is famously mysterious. Hence primary conclusion number 1 relies partly on a widely shared brute judgment about animal consciousness, not simply on science. There is in fact a defense of some kind of neo-Cartesian view in Mike Murray’s book _Nature Red in Tooth and Claw: Theism and the Problem of Animal Suffering_ (Oxford University Press, 2008). These reviews of Murray’s book and Trent Dougherty’s are good leads: https://ndpr.nd.edu/reviews/nature-red-in-tooth-and-claw-theism-and-the-problem-of-animal-suffering/
https://ndpr.nd.edu/reviews/the-problem-of-animal-pain-a-theodicy-for-all-creatures-great-and-small/
The author will need to do more work to say something new and interesting in light of this work.
Pace the author, it therefore does make sense to ask why the world is not different. Even granting that God wanted all the beaks, tails, hooves, claws, suckers, gills and wings precisely where and when they were (for whatever reason), why did He have to make it all hurt---when it would have been possible, indeed more natural by some standards, for it not to have felt like anything, good or bad? Do the animals’ conscious experiences cause their body parts to be in different places than they otherwise would have been (denial of epiphenomenalism)?
One might also ask why God bothered with evolution instead of just (so to speak) fast-forwarding to His favorite scene, the human era (if it indeed is His favorite). Does the evolutionary process's having really happened, rather than just being empirically adequate, achieve something important? Is some notion of divine veracity upheld? The author would do well to do more to justify scientific realism, a standard debate in the philosophy of science, rather than simply take it for granted. Indeed various secular philosophers of science have felt pain over their inability to give a good answer on this issue:
John D. Norton, “Observationally Indistinguishable Spacetimes: A Challenge for Any Inductivist,” in Gregory J. Morgan, editor, _Philosophy of Science Matters: The Philosophy of Peter Achinstein_. Oxford University Press (2011), pages 164-176.
John Earman, “Till the End of Time,” in John S. Earman, Clark N. Glymour, and John J. Stachel, _Foundations of Space-Time Theories_. University of Minnesota Press, Minneapolis (1977), pages 109-133. Minnesota Studies in the Philosophy of Science VIII.
John Byron Manchak, “What Is a Physically Reasonable Space-Time?” _Philosophy of Science_ 78 (2011) pages 410-420.
If there is no secular answer to this question, then one is probably relying on a theological answer, which might or might not be unique.
I appreciate the author’s drawing on Polish-language sources. On the other hand, if those sources also exist in English, then citing them in Polish seems less well motivated unless there is some specific reason. The Brothers Karamazov is of course well known. I wonder if Bochenski’s Logika Religii (1990) is a reprinting of the original of the 1965 NYU Press-published Logic of Religion?
Author Response
All detailed responses are in the file.

Reviewer 3 Report
Comments and Suggestions for Authors
General remark:
I suggest to refer (and discuss) more concepts in contemporary theodicy (e.g. R. Swinburne, A. Plantinga, J. Hick; from Poland – J. Å»yciÅ„ski, J. WoleÅ„ski).
Detailed remarks:
Par. 1:
„Darwin’s idea that evolution is guided by random mutations and natural selection”
Remark: Darwin knew the mechanism of natural selection, but he didn’t know the concept of the gene and the mechanism of mutation (this is „the synthetic theory of evolution” idea - first half of the 20th century).
-
Par. 2:
Remark: I suggest that phrase „picture of the world” to replace by „image of the world”.
-
Chapter 2:
„His conclusion was that the world is bad, but it is still the best of all possible worlds”
Remark: Leibniz didn’t directly suggest that the world is bad. He claimed, that the world is – despite the evil – the best of all possible (created as an effect of God’s thinking, pre-established harmony; see: Monadology).
-
Chapter 2:
„Epicurus presents a spectrum of three ways of conceiving of God in the face of evil”
Remark: Epicurus Dilemma leads to four possibilities:
a) God is neither able nor willing
b) God is able but not willing
c) God is unable and willing
d) God is able and willing
-
Chapter 3.2: I suggest to:
a) describe the concept of teleonomy (not the same as purposefulness; it means directionality of evolutionary processes, fulfillment of an evolutionarily programmed goal, although the entire evolutionary process is not aimed at a general purpose),
b) compare this idea to the concept of teleology.
Author Response
All detailed responses are in the file.

Round 2
Reviewer 1 Report
Comments and Suggestions for Authors
Second review of intelligibility paper
Despite the editing of the paper in response to referees, the key link between theodicy and intelligibility is still not made. Therefore the paper still does not arrive at a clear position on the theodicy of natural evil, and therefore is unable to interact in sufficient detail with existing positions in evolutionary theodicy. It does not make an original contribution to knowledge in the area.
Specific comments:
Lines 124-29: If this issue is introduced, it should be treated in more detail. But I suggest that the issue of what Darwin thought is not central to the argument.
Lines 299-301: This is quite confused and needs significant clarification.
Lines 312-315: It is not clear why the issue of realism is introduced. If it is to be discussed, it needs much more teasing out.
Lines 458-459: This is still too general a statement. Many complex physical processes are not about self-organisation.
Line 511: goal-directedness has not been demonstrated.
Line 558: suffering is inscribed into the process of life. This conclusion can be found in Schneider 2020, who should be cited here.
Line 558-563: The crucial theodical question is which of 1) and 2) is to be preferred. But the article does not tell us this. The paragraph that follows should help us but is not clear enough to do so. Are we to conceive of life only in a positive way, or not? At this crucial juncture in the argument, we are not offered a clear conclusion, nor is that conclusion critically compared with other positions in evolutionary theodicy.
Line 593: Should this be ‘to that of the Incarnation’?
Comments on the Quality of English Language
There are still some infelicities in the English, such as over-use of the definite article, which need rigorous copy-editing.
Author Response
All answers in the attachment.

Reviewer 2 Report
Comments and Suggestions for Authors
This paper draft is now better referenced, better argued, and more modest than before. The author has taken many of my suggestions on board. The author has at least gestured toward the possibility of neo-Cartesian responses to animal suffering; possibly that is all that such responses deserve, but I am glad for the mention. The author has also acknowledged the realism-antirealism debate’s relevance at least in passing, though likely without recognizing potentially far-reaching implications.
It is helpful to see the Track Changes-like format, but also unclear at times what the new wording is, with various bizarre forms like “thea” appearing repeatedly.
I wonder whether the enthusiastic, even fawning endorsement of Darwin, is useful---especially given that, as the author now admits, the problem of animal suffering does not even depend on whether animals change significantly over time, but only on the premise that they have been living and dying in ways rather like they do now for a long time and not plausibly due to Adam’s sin. It looks perhaps like an effort to strike a pose or cultivate a certain image.
I suggest that the author relies too heavily on the privation theory of evil. It isn’t a very plausible idea, and seems to be largely irrelevant to the real issues. (Todd C. Calder, “Is the Privation Theory of Evil Dead?” _American Philosophical Quarterly_ 44 no. 4 (Oct. 2007), pp. 371-381.) I also don’t think that someone deeply rooted in the Bible, especially Isaiah 45, should think that the privation theory of evil meets a real need. God doesn’t seem to be very worried about meeting our ethical standards. He also announces that He forms light and creates darkness, brings prosperity and creates disaster. Traditional Jews and Christians might be wary of trying to get God ‘off the hook’ in a way that God does not want.
Presumably the author did not intend to say that Copernicus paid homage to Galileo, given that Galileo was born after Copernicus’s death. That does seem to be implied by this passage, however: “From a historical point of view, the leading figures of the Scientific [188] Revolution (e.g., Copernicus, Gassendi, Boyle, Newton, Descartes) wished to avoid a clash [189] between theology and the new sciences by resorting to a view of the two Books, in homage [190] to Galileo, i.e., the Book of Nature and the Book of Scripture.”
Around line 360, I suggest that the author takes the idea of extrinsic intelligibility far too seriously. Linking it to effective technology (c.f. Torrance around line 427) suggests that extrinsic intelligibility is something that often exists. Admittedly Kant claims so, but so much the worse for Kant. Actual examples of extrinsic intelligibility would seem to be fairly trivial like finding faces or animals in cloud shapes, or finding the face of Jesus or Mary in a piece of toasted bread. If there are clear serious examples, it would be interesting to hear about them.
Around line 561, It would be good to see more argumentation. I am surprised by the claim that the conventional reading of Augustine is wrong; I have seen Augustine distorted or selectively received sufficiently often by people with modernizing agendas that more evidence is needed here. For example, people talk about proto-evolutionary ideas or his warning not to say stupid things about science, while ignoring his instantaneous creation, estimate of the age of the world from the genealogies in Genesis, and affirmation of waters above the heavens. Perhaps the author intends to call attention to congenial parts of Augustine while disagreeing with less congenial parts? In other words, it is not that people have traditionally misunderstood Augustine, but that many people today disagree with some passages while valuing others. My conjecture is that that is what is occurring here; if so, the author should say so, and if not, the author should show it.
Comments on the Quality of English LanguageThe English has suffered from the Track Changes format of the submission. It is not always clear which words are part of the submission.
Author Response
All answers in the attachment.
